# Genome-Wide Identification and Characterization of CDPK Gene Family in Cultivated Peanut (*Arachis hypogaea* L.) Reveal Their Potential Roles in Response to Ca Deficiency

**DOI:** 10.3390/cells12232676

**Published:** 2023-11-21

**Authors:** Shikai Fan, Sha Yang, Guowei Li, Shubo Wan

**Affiliations:** Institute of Crop Germplasm Resources, Shandong Academy of Agricultural Sciences, Ji’nan 250100, China; skfan09@126.com (S.F.); yangsha0904@126.com (S.Y.)

**Keywords:** cultivated peanut, calcium-dependent protein kinases, CDPK, gene expression, stress response, calcium deficiency

## Abstract

This study identified 45 calcium-dependent protein kinase (CDPK) genes in cultivated peanut (*Arachis hypogaea* L.), which are integral in plant growth, development, and stress responses. These genes, classified into four subgroups based on phylogenetic relationships, are unevenly distributed across all twenty peanut chromosomes. The analysis of the genetic structure of AhCDPKs revealed significant similarity within subgroups, with their expansion primarily driven by whole-genome duplications. The upstream promoter sequences of *AhCDPK* genes contained 46 cis-acting regulatory elements, associated with various plant responses. Additionally, 13 microRNAs were identified that target 21 *AhCDPK* genes, suggesting potential post-transcriptional regulation. AhCDPK proteins interacted with respiratory burst oxidase homologs, suggesting their involvement in redox signaling. Gene ontology and KEGG enrichment analyses affirmed *AhCDPK* genes’ roles in calcium ion binding, protein kinase activity, and environmental adaptation. RNA-seq data revealed diverse expression patterns under different stress conditions. Importantly, 26 *AhCDPK* genes were significantly induced when exposed to Ca deficiency during the pod stage. During the seedling stage, four *AhCDPKs* (*AhCDPK2/-25/-28/-45*) in roots peaked after three hours, suggesting early signaling roles in pod Ca nutrition. These findings provide insights into the roles of CDPK genes in plant development and stress responses, offering potential candidates for predicting calcium levels in peanut seeds.

## 1. Introduction

Plants have evolved complex mechanisms to adapt to their constantly changing environment. One such mechanism involves the use of calcium ions (Ca^2+^) as important cellular messengers, which play crucial roles in plant growth, development, and response to biotic and abiotic stress [1,2]. When external changes occur, the concentration of Ca^2+^ in the cytoplasm changes, triggering a series of physiological and biochemical reactions that improve plant tolerance to environmental stress [3].

To sense changes in Ca^2+^ concentration, plants have evolved various calcium-sensing proteins, including calmodulins (CaMs), calmodulin-like proteins (CMLs), calcineurin B-like proteins (CBLs), and Ca^2+^-dependent protein kinases (CDPKs) [4,5]. Among these, CDPKs are unique as they are Ser/Thr protein kinases capable of directly converting upstream Ca^2+^ signals into downstream protein phosphorylation events [5,6]. CDPKs are single peptide chains with four characteristic domains: a variable N-terminal domain (VNTD), a Ser/Thr protein kinase domain (PKD), an autoinhibitory junction region (JD), and a carboxyl-terminal calmodulin-like domain (CaMLD) [4,7]. The VNTD domain is poorly conserved and is involved in substrate recognition. Most CDPKs contain N-myristylation and/or S-palmitoylation sites, which enable the proteins to be localized to the membrane and contribute to the specificity of substrate recognition [8]. The PKD, which is relatively conserved in green plants, is responsible for binding ATP and phosphorylating downstream substrates [5,9,10]. The JD domain has kinase-inhibitory properties and can bind to the CaMLD domain. The CaMLD, located at the C-terminus, contains between one and four EF-hand motifs that bind to Ca^2+^ [9,10]. The differences in the number and amino acid sequences of these motifs enable CDPKs to decode different forms of Ca^2+^ signatures in the cytoplasm [7].

CDPKs play a crucial role in plant signaling, acting as key mediators of Ca^2+^ signals that are essential for various physiological responses [11]. CDPKs phosphorylate a range of substrates, including ion channels, transcription factors, and metabolic enzymes, which allow them to regulate diverse cellular processes [12]. As a result, CDPKs have been shown to play pivotal roles in a wide range of plant functions [7,11,12]. For example, *CDPK* genes are involved in plant growth and development [12]. In *Arabidopsis*, loss of *AtCPK2,6,20,17,34* reduced pollen tube growth, while *AtCPK11,24* increased [7,12]. The *MaCDPK7* gene is involved in regulating banana fruit ripening [13]. *TaCPK40* negatively regulates seed dormancy and positively regulates seed germination [14]. In addition, CDPKs have been shown to be involved in the response to abiotic stresses [7,11]. *CPK4* and *CPK11* overexpressing lines were more tolerant to drought stress through signal transduction in *Arabidopsis* [15]. Overexpression of *OsCDPK7* enhanced the induction of some stress-responsive genes in response to salinity/drought in rice [16]. Overexpression of *SiCDPK24*, *TaCDPK25-U-AS1*, or *TaCDPK25-U-AS2* enhanced drought resistance in *Arabidopsis* [17,18]. Overexpression of *GmCDPK3* improved drought and salt resistance in *Arabidopsis* [19]. *AtCPK28* phosphorylates and promotes the nuclear translocation of NIN-LIKE PROTEIN 7 (NLP7), thereby positively regulating plant response to cold stress in *Arabidopsis* [20]. The interaction between PpCDPK7 and PpRBOH may be the intersectional point of Ca^2+^ and ROS signal transmission during cold storage of peach fruits [21]. Genome-wide analyses have identified *CDPK* genes in various plant species, including *Arabidopsis thaliana* (34), rice (29), wheat (85), maize (40), soybean (50), cotton (84), chickpea (22), *Medicago truncatula* (24), peach (17), and pineapple (17) [6,17,21,22,23,24,25,26,27]. However, little is known about CDPKs in cultivated peanut (*Arachis hypogaea* L.).

Cultivated peanuts (*Arachis hypogaea*, allotetraploid, AABB, 2n = 4 × = 40) arose from the interspecific hybridization and subsequent chromosome doubling of two wild diploid ancestors, *Arachis duranensis* (AA genome, 2n = 2 × = 20) and *Arachis ipaensis* (BB genome, 2n = 2 × = 20) [28,29,30]. As an important economic and oil crop grown worldwide, peanuts often suffer from abiotic stress, such as drought, salt, and low temperatures [31]. In addition, it is important to note that peanut is a calcium-addicted crop, and calcium (Ca) deficiency can significantly impact the growth and development of the plant, resulting in various physiological changes [32,33]. Previous transcriptomic studies have indicated that Ca^2+^ deficiency in soil can lead to the abortion of peanut embryos or prevent kernel expansion, ultimately resulting in reduced peanut yields [34,35,36]. Certain CDPKs have been implicated in this process, although it is not yet clear if all CDPKs play a role in response to Ca deficiency. Therefore, it is necessary to comprehensively elucidate the role of the peanut CDPK gene family in response to Ca deficiency.

In our current study, we conducted a comprehensive analysis of the CDPK gene family in cultivated peanut (*AhCDPK*). These in silico analyses included genome-wide identification, characterization, genomic evolution, gene structure, conserved motifs, cis-regulatory elements, putative miRNA, protein-protein interactions, and functional annotations. Moreover, we also conducted expression profiling of *AhCDPK* genes in diverse tissues/organs, under abiotic stress, and Ca deficiency conditions using transcriptome and RT-qPCR techniques. By integrating these various bioinformatic and experimental approaches, we gained valuable insights into the evolutionary and functional roles of *AhCDPK* genes. Additionally, a comprehensive analysis of the response of peanut CDPK gene family members to Ca deficiency provided relevant information for elucidating the response mechanism of the CDPK gene family to the external Ca environment. These findings could have significant implications for further functional studies on the novel roles of *AhCDPK* genes in peanut breeding programs under challenging environmental conditions.

## 2. Materials and Methods

### 2.1. Identification of Peanut CDPK Genes

The genome file, protein file, coding sequences (CDS), and annotation files of *Arachis hypogaea*, *Arachis duranensis*, and *Arachis ipaensis* were obtained from Peanutbase (https://www.peanutbase.org/, accessed on 1 March 2023). The sequences of Arabidopsis isoforms were also obtained from TAIR (http://www.arabidopsis.org/, accessed on 1 March 2023). The sequences of rice isoforms were also obtained from RGAP (http://rice.uga.edu/, accessed on 1 March 2023). The Hidden Markov Model (HMM) was downloaded from the Pfam database in InterPro (https://www.ebi.ac.uk/interpro/, accessed on 1 March 2023). 

To identify homologs of CDPK proteins in peanut, a series of steps were taken. Firstly, 34 Arabidopsis CDPK proteins were used as query sequences in local BLASTP searches against the peanut protein genome. Then, the HMMER v3.3.2 program was employed to search for protein kinase domains (PF00069) and EF-hand domains (PF13499). To further verify the reliability of these candidate sequences, motif scanning of the CDPKs was performed using three databases, Pfam, PROSITE profiles, and SMART in the InterPro (https://www.ebi.ac.uk/interpro/, accessed on 6 March 2023), to confirm each candidate AhCDPK protein as a member of the AhCDPK family [37]. The EF-hand motif was analyzed using the InterPro database. In addition to the identification of CDPK homologs in wild diploid peanut, similar approaches were applied to diploid parents. In *A. duranensis*, AdCDPK1–AdCDPK22 was discovered, while in *A. ipaensis*, AiCDPK1–AiCDPK23 were identified.

GPS-Lipid 1.0 was employed to predict N-terminal myristoylation sites, while GPS-Palm was used to predict palmitoylation sites. Default settings and a high threshold were utilized in these predictions [38]. An analysis of physicochemical properties such as molecular weight (MW), isoelectric point (pI), and amino acid length of the finalized set of *AhCDPK* genes was performed using a local Perl script. Subcellular localization prediction was conducted using the BUSCA online program (https://busca.biocomp.unibo.it, accessed on 10 March 2023) [39]. The positions of *AhCDPK* genes on chromosomes were mapped using TBtools [40].

### 2.2. Phylogenetic, Conserved Motifs, Gene Structure, and Protein Tertiary Structure Analysis of AhCDPKs

Multiple alignments of CDPK protein sequences from peanut, *Arabidopsis*, and rice were performed using the MUSCLE program, with default parameters implemented in MEGA11 (https://www.megasoftware.net/, accessed on 27 August 2023) [41]. The phylogenetic tree was generated using MEGA11 software, employing the neighbor-joining method. The tree was constructed with 1000 bootstrap replicates using the Poisson correction method. Default parameters were utilized for all other settings during the tree construction process. In addition, the phylogenetic tree was displayed and modified using EvolView(https://evolgenius.info//evolview-v2/, accessed on 5 April 2023) [42]. The conserved motifs of each protein were analyzed using the MEME program (version 5.5.1) (http://meme-suite.org/tools/meme, accessed on 10 April 2023) [43]. The maximum motif number was set as 10, and the maximum and minimum motif length were set as 15 and 300, respectively, the other parameters were set as default. Phylogenetic tree and motifs protein structures were visualized using TBtools [40]. The gene structures were visualized with TBtools using the CDS and genomic sequences of *AhCDPK* genes [40]. The protein tertiary structure of AhCDPKs was predicted by the SWISS-MODEL server (https://swissmodel.expasy.org/, accessed on 28 July 2023). 

### 2.3. Gene Duplication Events and Synteny Analysis

The MCScanX toolkit was used to investigate gene duplication events, which were visualized using TBtools [40,44]. The synteny relationship of orthologous *CDPK* genes among peanut, rice, and *Arabidopsis* was shown using TBtools v1.098747 [40]. The synonymous and non-synonymous substitution rates (Ka = number of nonsynonymous substitutions/nonsynonymous sites; Ks = number of synonymous substitutions/synonymous sites) were computed using TBtools software v1.098747 [40]. The Ks value was used to calculate the divergence times of the duplication event (T = Ks/2λ), and the neutral substitution rate (λ) is estimated to be 8.12 × 10^−9^ for peanut [30]. 

### 2.4. Cis-Acting Regulatory Elements in Promoters and miRNA Target Predictions

The 2000 bp 5′-upstream sequences from the translation start site of all *AhCDPKs* were extracted by local Perl script and were used for the prediction of cis-acting regulatory elements according to PlantCARE (http://bioinformatics.psb.ugent.be/webtools/plantcare/html, accessed on 20 April 2023) [45], and then the results were visualized with ggplot2 (v3.4.4) in R language (v4.1.2).

MicroRNA target sites were analyzed using the psRNATarget database (https://www.zhaolab.org/psRNATarget/, accessed on 20 March 2023) [46]. The interaction network among miRNAs and *AhCDPKs* used Cytoscape (v3.9) [47]. 

### 2.5. Analysis of Protein Interaction Network and Functional Annotation Evaluation

To analyze the protein interaction network, the protein sequences of AhCDPKs were mapped to UniProtKB IDs using the protein interaction network prediction website (https://cn.string-db.org/cgi/input?sessionId=bzbREExWQRsE&input_page_show_search=on, accessed on 30 March 2023). *Arachis hypogaea* was selected as the organism to retrieve the protein network interactions map. The minimum required interaction score was set to 0.700, while default settings were used for the other parameters [48].

To perform Gene Ontology and KEGG annotation evaluations, AhCDPK protein sequences were submitted to eggNOG-mapper (http://eggnog-mapper.embl.de/, accessed on 5 May 2023) [49]. GO and KEGG enrichment evaluations were performed using TBtools [40]. The GO_Level was configured to range from three to eight, and both the *p*-value and corrected *p*-value (using the Benjamini-Hochberg method) were set to be less than 0.01.

### 2.6. RNA-Seq Data Analysis

The published transcriptome data (NCBI BioProject no. PRJNA291488 (one, two, or three replicates), PRJNA470988 (three replicates), PRJNA657965 (three replicates), PRJNA751249 (three replicates), PRJNA560660 (three replicates)) of peanut were used to analyze *AhCDPKs* expression patterns (Appendix A). The cultivated peanut Tifrunner was used as a reference genome (gnm2. J5K5, https://www.peanutbase.org/genome/, accessed on 1 March 2023), and the RNA-seq data were analyzed using fastp, Hisat2, samtools, and edgeR to obtain TPM (Transcripts Per Kilobase Million) values [50,51,52,53]. The average of all replicates for each treatment was calculated, and the resulting data were visualized using heatmaps generated using TBtools [40]. For the stress treatment, the transcript levels after treatments were rescaled relative to that untreated time (at 0 h) when calculating the relative expression levels. For Ca deficiency treatment, the transcript levels after treatments were rescaled relative to control (Ca sufficiency) when calculating the relative expression levels. For all treatments, the heatmap portrayed the relative expressions after the log2 transformation. Colors from blue to red represent relative expression levels from low to high in every heatmap.

### 2.7. Plant Growth and Treatments

The peanut cultivar Fenghua 1 was used in this study. Seedlings of peanut were grown in 1/2 Hoagland solution in a light incubator under a 16 h photoperiod (32 °C) and 8 h dark (25 °C) period. Samples were collected from different tissues of roots, leaves, and stems at the seedling stage after two weeks of growth. For Ca deficiency treatment, two-week-old seedlings were subjected to a Ca-free nutrient solution (without Ca(NO_3_)_2_), and nitrogen (N) was balanced using NH_4_NO_3_. Root and leaf samples were collected at 3, 6, 12, and 24 h after treatment, with non-treated samples at 0 h serving as the control. For the stress treatment, two-week-old seedlings were exposed to drought (20% (m/V) PEG6000), salt (150 mM NaCl), and cold (4 °C). Leaf samples were collected at 3, 6, 12, and 24 h after treatment, with non-treated samples at 0 h serving as the control. All treatments were replicated three times. The samples were immediately frozen in liquid nitrogen and then stored at −80 °C until RNA extraction.

### 2.8. RNA Extraction and qRT-PCR Analysis

Total RNA was extracted using a FastPure Cell/Tissue Total RNA Isolation Kit V2 (Vazyme, Nanjing, China), and cDNA synthesis was performed using PrimeScript™ RT reagent Kit (Takara, Dalian, China) according to the manufacturer’s instructions. The Taq Pro Universal SYBR qPCR Master Mix (Vazyme, Nanjing, China) was used to conduct qRT-PCR amplification in the 7500 Fast Real-time PCR System (Applied Biosystems, Foster City, CA, USA). All primers were designed using the DNAMAN 9 software (Lynnon Corporation, Quebec, QC, Canada). The sequence information of all primers is listed in Appendix A. All the genes tested resulted in a single melt peak, confirming that the primers used were appropriate and specific for amplifying the desired target gene (Appendix A). Two technical replicates were performed for each treatment. The relative expression level was calculated using the 2^−ΔΔCt^ method [54]. For Ca deficiency and stress treatments, the expression of 0 h in each treatment was considered “1”. All expression data were means ± SD (n = 6). Means were compared using Duncan’s multiple range test at *p* < 0.05 in all cases in R version 4.1.2 and different letters represent significantly different values. Gene expressions were visualized using GraphPad Prism v9.5.0 (GraphPad Software, Boston, MA, USA).

## 3. Results

### 3.1. Genome-Wide Identification of 45 CDPK Genes in Peanut Genome

In this study, genome-wide analysis of the CDPK gene family has been performed on *A. hypogaea* genome sequence using profile HMM searches and BLASTP. A total of 45 CDPK family genes were identified in peanut after confirming each candidate using Pfam, ScanProsite, SMART, and InterProScan to verify the EF-hand and Pkinase-conserved domain. We renamed these genes *AhCDPK1* to *AhCDPK45* according to their chromosome positions (Appendix A). The chromosomes of the *A. hypogaea* genome are numbered Arahy.01–Arahy.20 [29]. By chromosomal localization analysis, 45 *AhCDPK* genes were determined to be unevenly distributed across the 20 chromosomes and mostly concentrated at both ends of the chromosomes (Figure 1). Additionally, 23 and 22 of them belonged to the A genome (Arahy.01–10) and B genome (Arahy.11–20), respectively. The number of *AhCDPKs* distributed on Arahy.02 is the largest, with a total of four. Arahy.03, 04, 05, 07, 12, 13, 14, 15, and 17 all have three *AhCDPKs*. Arahy.01, 06, 11, and 16 all have two *AhCDPKs*. The number of *AhCDPKs* distributed on Arahy.08, 09, 10, 18, 19, and 20 is the least, with only one for each. As shown in Appendix A, the gene length of *AhCDPKs* varies from 3335 bp (*AhCDPK7*) to 12,525 bp (*AhCDPK33*), with CDS length varying from 1425 bp (*AhCDPK41*) to 2325 bp (*AhCDPK38*). The protein lengths also vary considerably, ranging from 475 (*AhCDPK41*) to 774 (*AhCDPK38*) amino acids, and the predicted MW ranged from 53.59 (*AhCDPK41*) to 87.95 (*AhCDPK38*) kDa, while the isoelectric point (pI) ranged from 5.03 (*AhCDPK3*) to 9.53 (*AhCDPK35*). Most AhCDPKs contain both predicted N-terminal myristoylation and palmitoylation sites, while AhCDPK8, 17, 31, and 38 contain only a predicted myristoylation site. AhCDPK21, 22, 43, and 44 were predicted to have neither myristoylation nor palmitoylation sites. Next, the subcellular localizations of AhCDPKs were predicted, and the results indicate that 28 members were localized in the nucleus, 15 in the chloroplast, and 2 in the cytoplasmic membrane. In addition, 22 genes (*AdCDPK1*–*AdCDPK22*) from *A. duranensis* and 23 genes (*AiCDPK1*–*AiCDPK23*) from *A. ipaensis* genomes were also recognized to study the evolution of *CDPK* genes between tetraploid and diploid parents (Appendix A).

### 3.2. Synteny Analysis of CDPK Genes in Peanuts

Gene duplication is indeed a crucial mechanism for the amplification of large gene families in plants, including peanut. The presence of 45 *AhCDPK* genes in peanut suggests that the CDPK gene family has undergone significant expansion during evolution. We next examined the segmental or tandem duplication events of the CDPK gene family in the peanut genome. Almost all the *AhCDPK* genes experienced gene duplication events except *AhCDPK3* and *AhCDPK29*, resulting in 47 replication events, including 33 pairs of whole-genome duplications (WGDs), 12 pairs of segmental duplications, and 2 pairs of tandem duplications (Figure 2), indicating that WGDs were the main way for expanding *AhCDPK* genes in peanut. We also calculated the synonymous (*Ks*), nonsynonymous (*Ka*), and *Ka*/*Ks* ratio values for each duplication event to study the evolutionary relationship among *AhCDPK* genes (Appendix A). The results showed that the *Ka* values of WGDs, segmental duplications, and tandem duplications ranged from 0 to 0.2980, from 0.0476 to 0.2791, and from 0.2312 to 0.2377, respectively, while the *Ks* values ranged from 0 to 2.0362, from 0.6378 to 1.3353, and from 0.5499 to 0.5647, respectively. The *Ka*/*Ks* ratio (0–0.4322) was always less than 1 for every *AhCDPK* gene pair (Figure 2, Appendix A), indicating that the *AhCDPK* gene family experienced strong negative selection. The divergence time for the 33 WGD pairs ranged from 0 to 125.3836 million years ago (Mya). The segmentally duplicated gene pairs had a divergence time ranging from 39.2710 to 82.2226 Mya. As for the two tandem duplicated genes, their divergence times were 33.8620 and 34.7708 Mya. 

To further study the potential evolution of the *AhCDPK* gene family, two comparative syntenic maps of peanut with four species, including its two diploid ancestors (*A. duranensis* and *A. ipaensis*) and two other dicot species (*Arabidopsis* and rice), were built to clarify the syntenic relationships of *CDPK* genes in peanut (Figure 3). Syntenic relationships were observed between 37 *AhCDPK* genes and 21 *AdCDPK* genes in *A. duranensis*, 38 *AhCDPK* genes and 21 *AiCDPK* genes in *A. ipaensis*, 22 *AhCDPK* genes and 16 *AtCDPK* genes in *Arabidopsis*, and 10 *AhCDPK* genes and 7 *OsCDPK* genes in rice. The number of orthologous pairs between peanut and these four species (*A. duranensis*, *A. ipaensis*, *A. thaliana*, and rice) was 55, 56, 33, and 11, respectively (Appendix A).

### 3.3. Phylogenetic Tree, Conserved Motifs, Gene Structure, and Tertiary Structures of AhCDPKs

To better elucidate the evolutionary relationship of AhCDPKs, a neighbor-joining (NJ) phylogenetic tree was constructed based on 153 CDPK protein sequences from peanut (red star), *Arabidopsis* (green triangle), rice (blue square), *A. duranensis* (dark green circle), and *A. ipaensis* (dark red circle) (Figure 4). CDPKs of peanut and its two diploid ancestors (*A. duranensis* and *A. ipaensis*) are more clustered together within the same branch than *Arabidopsis* and rice, showing a closer genetic relationship. According to the recognized classifications of the *Arabidopsis* and rice CDPK families [10,22], the AhCDPK/AdCDPK/AiCDPK proteins were divided into four subgroups, including 15/8/8, 14/6/7, 12/6/6, and 4/2/2 members in Subgroup I, II, III, and IV, respectively. 

Phylogenetic subgroups were often related to genes and protein structures. The presence of four subgroups within the 45 AhCDPK proteins indicates that the gene family has undergone conservation within these subgroups. To illustrate the structural characteristics of the 45 CDPKs from peanut, firstly, we performed a motif and domain analysis combining the MEME program and Pfam, SMART databases, and PROSITE profiles in the InterPro (Figure 5). All AhCDPK proteins contained the four typical domains, VNTD, PKD, JD, and CaMLD. A total of 10 conserved motifs with lengths ranging from 21 to 105 amino acids were identified using MEME. (Figure 5A, Appendix A). All proteins were found to contain motifs 1, 2, 3, 4, 5, and 6, while some motifs were detected only in specific subgroups. For example, motif 7 was found only in Subgroup III, motif 10 was not found in Subgroup IV, and motif 18 was found only in Subgroup I. Motif 9 was found only in AhCDPK5/-11/-13/-19/-23/-27/-33/-35/-42/-45. PKD showed conserved motifs 1, 3, 4, 5 in MEME, and the protein kinase domain in Pfam, serkin_6 in SMART, and the protein kinase domain profile in PROSITE were observed in the domain (Figure 5B–D). CaMLD were observed in motif 2 in MEME, EF-hand domain pair in Pfam, efh_1 in SMART, and EF-hand calcium-binding domain profile in PROSITE. The JD existed in motif 6, and motif 8 was found in VNTD, although nine proteins had none. All the AhCDPKs contained four EF-hands except AhCDPK41, which possessed three. Overall, the protein structure of this family was relatively conserved.

Then, the gene structure was analyzed. All peanut CDPK genes contain introns, and the number of introns ranges from 6 to 11 (Figure 6). There were 12, 18, 7, 3, and 5 genes containing 6, 7, 8, 9, and 11 introns, accounting for 26.67%, 40%, 15.56%, 6.67%, and 11.11% of the total genes, respectively. Notably, all the genes in Subgroup IV had the most at 11 introns, which showed highly similar genetic structures. In addition, there were 6 to 8 introns in Subgroup I, 7 to 11 introns in Subgroup II, and 6 to 9 introns in Subgroup III, respectively. By combining the gene structure of *AhCDPKs* with the phylogenetic tree, this result revealed that the exon-intron distribution of *AhCDPKs* was related to its classification.

Protein tertiary structures provide valuable insights into the structural characteristics of proteins. To further illustrate the structural characteristics, the tertiary structures of the CDPK proteins in peanut were investigated using AlphaFold2 as it is a novel machine learning approach and can predict protein structures with experimental accuracy based solely on their primary amino acid sequences [55]. The tertiary structures of the AhCDPK proteins were modeled using 19 template proteins identified through AFDB searches (Figure 7). The resulting models showed high quality, with GMQE values ranging from 0.666 to 0.830, sequence identities between 72.66% and 100%, and coverage from 88.1% to 100% for nearly the full length of each protein (Appendix A). In addition, there were 6/7/5/1 templates in Subgroup I/II/III/IV, respectively. These templates were found to match 1, 2, or 4 proteins in each subgroup, with similar structures indicating a conserved function within each subgroup. Furthermore, transmembrane structure analysis showed that none of the AhCDPK proteins contained a transmembrane domain (Appendix A). In summary, by examining conserved motif structures, gene structures, protein tertiary structures, and phylogenetic relationships, it is evident that the gene organizations within subgroups of CDPK proteins remain remarkably consistent. This indicates that CDPK proteins possess highly conserved amino acid residues, and CDPK members belonging to the same subgroup likely perform similar functions.

### 3.4. Cis-Regulatory Elements Prediction in the Promotors of AhCDPKs

Considering the relatively conserved structures present within the CDPK gene family, we initially focus on exploring the potential regulatory factors affecting *AhCDPK* gene expression to better understand the functions of this gene family. Thus, the recognized putative cis-acting regulatory elements were analyzed from 2 kb upstream of the coding sequence of 45 *AhCDPK* genes using PlantCARE (Figure 8, Appendix A). As shown in Figure 8, a total of 46 unique cis-acting regulatory elements with numbers equal to or greater than 10 were identified in the promoters of *AhCDPKs*, which were divided into the following five main categories: (1) phytohormone responsive, including ABA-responsive element (ABRE, ABRE3a, and ABRE4), auxin-responsive element (TGA-element), ethylene-responsive element (ERE), MeJA-responsive element (TGACG-motif, CGTCA-motif), GA-responsive element (P-box), and salicylic acid-responsive element (as-1, TCA, and TCA-element) (Appendix A); (2) abiotic stress, including anaerobic induction element (ARE), low-temperature-responsive element (LTR), drought-inducibility element (MBS), dehydration-responsive elements (Myb, MYB, Myb-binding site, MYB-like sequence, Myc, and MYC), stress-responsive element (STRE), and defense and stress-responsive element (TC-rich repeats) (Appendix A); (3) biotic stress, including wound-responsive element (WRE3, WUN-motif) and W box (Appendix A); (4) light responsive, including AE-box, AT1-motif, ATCT-motif, Box 4, GATA-motif, G-box, G-Box, GT1-motif, I-box, MRE, TCCC-motif, and TCT-motif (Appendix A); (5) plant growth and development, including secondary xylem development element (AAGAA-motif), A-box, maximal elicitor-mediated activation element (AT-rich sequence), meristem-expression element (CAT-box, CCGTCC-box, CCGTCC-motif), endosperm-expression element (GCN4-motif), zein metabolism regulation element (O2-site), and seed-specific regulation element (RY-element) (Appendix A). Out of the five main categories, abiotic stress had the highest number of cis-acting regulatory elements, with a total count of 603. The remaining categories, in descending order, were light responsive (546), phytohormone responsive (505), plant growth and development (155), and biotic stress (79). Among the analyzed genes, *AhCDPK43* had the highest number of cis-acting regulatory elements within its promoter region (87), as it contained two prominent elements, ABRE and G-box, with a remarkable count of nineteen each (Figure 8A,B). Box 4 (197), ERE (127), MYC (121), MYB (115), and ABRE (114) were identified as the top five most prevalent cis-acting regulatory elements detected in the promoter regions of most genes (Figure 8C), indicating that *AhCDPKs* play an essential role in response to phytohormone, light, and abiotic stresses.

### 3.5. Prediction of miRNAs Targeting AhCDPK Genes

MicroRNAs (miRNAs) are crucial players in post-transcriptional regulation. To further investigate the potential regulatory mechanism in the regulation of *AhCDPK* genes, we identified a total of 13 miRNAs that target 21 genes. These miRNAs belonged to eight different families. The results revealed that ahy-miR3520-3p targeted the highest number (5) of genes (Figure 9, Appendix A). Four miRNAs, including ahy-miR156a, ahy-miR156c, ahy-miR3511-3p, and ahy-miR3520-5p, targeted four genes, followed by ahy-miR156b-5p targeting three genes. Moreover, five miRNAs, including ahy-miR3510, ahy-miR3513-3p, ahy-miR3513-5p, ahy-miR3514-5p, and ahy-miR408-3p, targeted two genes each. Additionally, ahy-miR167-5p and ahy-miR3514-3p targeted one gene each. Interestingly, all four genes in Subgroup IV were targeted by miRNAs, while half of the genes in Subgroup II were targeted by miRNAs. One-third of the genes in both Subgroup I and Subgroup III were also targeted by miRNAs. These results suggest that miRNAs may play a crucial role in the posttranscriptional regulation of *AhCDPK* genes and their expression levels. 

### 3.6. Protein Interaction Network and Functional Annotation Analysis of AhCDPKs

After analyzing the regulatory factors at the gene level, we next investigated potential regulation at the protein level. Firstly, an AhCDPKs protein interaction network was constructed between AhCDPKs and other peanut proteins using the STRING database [48]. According to the predicted results, we identified 32 AhCDPKs that interacted with 5 different peanut proteins (Figure 10, Appendix A). Among these AhCDPKs, both AhCDPK22 and AhCDPK43 were found to interact with AhCDPK7 and AhCDPK30, respectively. Interestingly, AhCDPK30 was found to interact with AhCDPK21 and AhCDPK8, while AhCDPK21 was found to interact with AhCDPK7, and AhCDPK8 was found to interact with AhCDPK7. It is noteworthy that all 32 AhCDPK proteins were found to interact with five other peanut proteins, namely A0A444Z8M2, A0A444ZZ96, A0A445A176, A0A445AZG3, and A0A445DW03. These peanut proteins were identified as AhRBOHF (respiratory burst oxidase homolog F), AhRBOHI, AhRBOHE, AhRBOHB, AhRBOHA, which are homolog genes of *Arabidopsis thaliana*, owing to the incomplete annotation of the peanut genome. The above five peanut proteins identified in our investigation all belong to the RBOH family. RBOHs, also known as NADPH oxidases, are integral plasma membrane proteins that play a critical role in the generation of reactive oxygen species (ROS) [56]. These results provided valuable information for the further functional characterization of AhCDPK proteins. Next, we investigated the role of AhCDPK proteins at the molecular level through gene ontology (GO) and KEGG enrichment analysis (Figure 11, Appendix A). To ensure the accuracy and reliability of GO and KEGG enrichment evaluations, the GO level was set to 3–8, and both the *p*-value and corrected *p*-value (BH method) were set to less than 0.01. The GO enrichment analysis can be divided into three main categories: biological processes (BP), molecular functions (MF), and cellular components (CC). For instance, in MF class, 29 enriched terms were detected, namely, calcium ion binding (GO:0005509), protein binding (GO:0005515), ion binding (GO:0043167), anion binding (GO:0043168), cation binding (GO:0043169), kinase activity (GO:0016301), calmodulin-dependent protein kinase activity (GO:0004683), calcium-dependent protein serine/threonine kinase activity (GO:0009931), protein kinase activity (GO:0004672), calcium-dependent protein kinase activity (GO:0010857), calmodulin binding (GO:0005516), protein serine/threonine kinase activity (GO:0004674), etc. In CC class, 9 enriched terms were detected, namely, nucleus (GO:0005634), cytoplasm (GO:0005737), intracellular membrane-bounded organelle (GO:0043231), membrane-bounded organelle (GO:0043227), intracellular organelle (GO:0043229), organelle (GO:0043226), plasma membrane (GO:0005886), intracellular anatomical structure (GO:0005622), cell periphery (GO:0071944). Whereas in BP class, 29 enriched terms were detected, including cellular response to stimulus (GO:0051716), signaling (GO:0023052), regulation of cellular process (GO:0050794), phosphorus metabolic process (GO:0006793), protein autophosphorylation (GO:0046777), intracellular signal transduction (GO:0035556), protein phosphorylation (GO:0006468), phosphorylation (GO:0016310), signal transduction (GO:0007165), regulation of response to stress (GO:0080134), regulation of response to stimulus (GO:0048583), response to osmotic stress (GO:0006970), response to abscisic acid (GO:0009737), response to salt stress (GO:0009651), etc. In addition, the KEGG pathway enrichment study identified six pathways involved in the different functions of the *AhCDPK* genes. The highly enriched pathways include protein kinases (01001), environmental adaptation (B09159), organismal systems (A09150), protein families: metabolism (B09181), plant-pathogen interaction (04626), and brite hierarchies (A09180). Overall, the GO and KEGG enrichment analyses validate the functional contribution of AhCDPK proteins in several important processes, including calcium ion binding, calmodulin-dependent protein kinase activity, regulation of cellular process, response to stresses, and environmental adaptation. 

### 3.7. Spatial Expression Analysis of AhCDPKs

The evidence from *cis*-regulatory elements, miRNA, protein interaction networks, and functional annotations underscores the importance of AhCDPKs in plant development (Figure 8, Figure 9, Figure 10 and Figure 11). To gain insights into the potential biological functions of AhCDPKs in growth and development, expression pattern analysis is a useful tool. In this study, we analyzed the expression levels of AhCDPKs in 22 representative tissues using public transcriptome data [57]. A heatmap was created through the hierarchical clustering of the gene expression profiles of 45 *AhCDPK* genes, and these could be divided into three clusters: Cluster A, B, and C, including 17, 12, and 16 members, respectively (Figure 12). Cluster A’s *CDPK* genes were highly expressed in most of the 22 different tissues, implying that these genes may play important roles in peanut development. However, most of Cluster B’s *CDPK* genes were lowly expressed in the 22 tissues except in the lateral stem leaf, in which *AhCDPK7/-30/-32* were highly expressed, suggesting that they may be involved in the growth and development of the organ. Notably, *AhCDPK10/-32* had the highest expression level in the lateral stem leaf, while lightly even not expressed in other tissues, indicating their role in specific development processes. A considerable number of Cluster C’s *CDPK* genes exhibited moderate expression levels across the 22 tissues. Among Cluster C’s *CDPK* genes, *AhCDPK45* was not expressed in the vegetative shoot tip or reproductive shoot tip but was highly expressed in the pattee 1 pod and pattee 10 seed. *AhCDPK23* was not expressed in the vegetative shoot tip or the lateral stem leaf. The above results indicated that *AhCDPK* genes showed diverse expression patterns in peanut.

In order to further validate the role of *AhCDPK* genes in development, we conducted expression profiling of 12 genes, with 4 genes selected from each cluster, in the roots, leaves, and stems of peanut seedlings (Figure 12B). The results of RT-qPCR analysis confirmed a similar trend to that observed in the RNA-seq data. In Cluster A, the expression levels of *AhCDPK8/-31/-37-40* were found to be the highest among all three tissues (roots, leaves, and stems). This indicates that these genes may play a significant role in the development of peanut seedlings. In Cluster B, the expression levels of *AhCDPK12/-13/-35/-42* were relatively high, suggesting their involvement in developmental processes as well. However, we observed that the expression levels of *AhCDPK7/-10/-30/-32* were hardly detected or had very low values in all three tissues. This suggests that these genes may have limited or specific functions in the development of peanut seedlings. This example demonstrates the consistency between the expression profiles obtained from RT-qPCR analysis and the RNA-seq data for all three clusters. The expression patterns of the selected *AhCDPK* genes in different tissues provide further evidence of their involvement in peanut development.

### 3.8. Expression Analysis of AhCDPKs under Abiotic Stresses

During growth and development, plants are significantly influenced by various environmental conditions, including low temperatures, high salinity, and drought. To investigate the potential biological functions of *AhCDPKs* in abiotic stress, we also analyzed the expression profiles of peanut CDPK genes in seeding responding to three abiotic stresses (salt, drought, and cold) using three public transcriptomes. As there were very low even zero TPM values in *AhCDPK10/-27/-32/-41* under control conditions, the other 41 genes were considered in the following analysis. Following a 16 h salt treatment, *AhCDPK5/-7/-18* were significantly induced (>2-fold increase from 0 h), while *AhCDPK11/-13/-23/-33* were repressed (>2-fold decrease from 0 h) (Figure 13A). Under drought treatment, *AhCDPK5* transcripts were upregulated after 4, 8, and 12 h, peaking at 4 h, whereas *AhCDPK11/-33* transcripts were downregulated at all three time points. *AhCDPK18* transcripts were upregulated after 8 and 12 h, whereas *AhCDPK31* transcripts were downregulated at both time points. *AhCDPK7/-17/-28/-38* transcripts were increased only after 4 h, whereas *AhCDPK12/-34* transcripts were decreased only at this time point. *AhCDPK2/-25* transcripts were increased only after 8 h, whereas *AhCDPK4/-23/-30* transcripts were decreased only at this time point. Only *AhCDPK8* transcripts were decreased after 12 h (Figure 13A). Under cold treatment, *AhCDPK2/-5/-7/-17/-18/-25* transcripts were upregulated after 24 and 48 h, while *AhCDPK22/-44* transcripts were downregulated at both time points. *AhCDPK1/-23/-24/-45* transcripts were increased only after 3 h. *AhCDPK38* transcripts were increased only after 24 h, while *AhCDPK12/-34* transcripts were decreased only at this time point. *AhCDPK8/-31* transcripts were decreased only after 48 h (Figure 13A). It is noteworthy that *AhCDPK5/-7/-18* were upregulated by cold, drought, and salt stresses, while *AhCDPK2/-17/-25/-38* were upregulated by cold and drought stress. On the other hand, *AhCDPK11/-33* were downregulated by drought and salt stresses, while *AhCDPK8/-12/-31/-34* were downregulated by cold and drought stresses. These results indicate that *AhCDPKs* may play an important role in response to various abiotic stresses, and different genes exhibit specific responses to stress.

To further assess the role of *AhCDPKs* in abiotic stresses, *AhCDPK5/-7/-18* were selected to investigate their detailed expression patterns using an RT-qPCR analysis. Leaves from peanut seedlings were collected, and the expression levels of these genes were measured at different time points: 3 h, 6 h, 12 h, and 24 h (Figure 13B–D). Under drought treatment, the transcripts of *AhCDPK5* were increased at all four time points, with the highest expression level observed at 3 h. *AhCDPK7* transcripts were increased at 3 h and 6 h. *AhCDPK18* transcripts showed a gradual increase over time, with more significant inductions at 12 h and 24 h. When subjected to salt treatment, *AhCDPK5* transcripts were significantly induced at all four time points, with the peak expression observed at 12 h. *AhCDPK7* transcripts were increased at all four time points. *AhCDPK18* transcripts were also increased at three time points, except for the 3 h time point. Under cold treatment, *AhCDPK5* transcripts were increased at three time points and showed a gradual increase over time. *AhCDPK7* transcripts were increased at 12 h and 24 h. *AhCDPK18* transcripts showed an increasing trend over time and peaked at 24 h. These RT-qPCR expression results were consistent with the RNA-seq results and highlighted the importance of *AhCDPK5/-7/-18* in response to abiotic stress. 

### 3.9. Expression Analysis of AhCDPKs in Response to Ca-Deficiency

Nutrient deficiency can greatly affect plant growth and development. For peanut, Ca is considered the third essential nutrient, following nitrogen and phosphorus. It has been reported that certain CDPKs are involved in seed and pod development when there is a deficiency of free Ca^2+^ in soil [33,58]. In order to explore the potential regulatory mechanisms of all *AhCDPKs* in the development process, the expression patterns of 43 *AhCDPKs* were analyzed. This was conducted using published RNA-seq data of Ca-deficient and Ca-sufficient peanut pods at 15, 20, and 30 days after pegging [36]. *AhCDPK10/-32* were excluded from the analysis due to their limited expression in most pod processes. As shown in Figure 14A, 26 *AhCDPK* genes were raised to a significant level (>2-fold) by at least one Ca deficiency treatment, accounting for 60% of all analyzed genes. Out of these 26 genes, *AhCDPK1/-2/-6/-23/-24/-25/-28/-45* exhibited >4-fold increase in all three Ca deficiency treatments, in comparison to the control. *AhCDPK9/-40* soared to an elevated level (>2-fold) after 15 d and >4-fold after both 20 d and 30 d Ca deficiency treatments. In addition, *AhCDPK15/-16/-17/37* were raised to a significant level (>2-fold) across all three Ca deficiency treatments. Similarly, *AhCDPK7/-19/-38/-39/-42* genes were upregulated (>2-fold) in two Ca deficiency treatments, while *AhCDPK3/-5/-8/-11/-13/-31/-33* were upregulated (>2-fold) in only one Ca deficiency treatment. On the contrary, *AhCDPK41* experienced a significant down-regulation (>4-fold) via 15d Ca deficiency treatment and >2-fold under 30 d Ca deficiency treatments. Furthermore, *AhCDPK27* was solely downregulated (>2-fold) by the 20 d Ca deficiency treatment. The above results suggest that *AhCDPK* genes may have specific roles in the adaptive response of peanut plants to Ca deficiency. 

To comprehensively explore the involvement of the CDPK gene family in response to Ca deficiency, we undertook a detailed investigation of the expression patterns of eight *AhCDPKs* that were significantly induced by Ca deficiency. Specifically, we employed qRT-PCR to analyze the expression levels of these *AhCDPKs* at different time points (3 h, 6 h, 12 h, and 24 h) in both the leaves and roots of peanut seedlings (Figure 14B,C). The results from our analysis of leaves indicated that the transcript levels of *AhCDPK1/-2/-6/-23/-24/-45* were downregulated at all four time points, while *AhCDPK28* was downregulated at 3 h and 12 h, but upregulated at 24 h. Interestingly, *AhCDPK25* was upregulated at 3h, 12h, and 24 h, providing us with a glimpse into the complexity of the molecular response mechanisms of plants. However, in the case of roots, we observed at least one upregulated time point of these eight *AhCDPKs*. Specifically, *AhCDPK1* was upregulated at 6 h, *AhCDPK2/-45* were upregulated at 3 h, *AhCDPK6* was upregulated at 24 h, *AhCDPK23/-24* were upregulated at 12 h and 24 h, and *AhCDPK25/-28* were upregulated at all four time points. Notably, we discovered that four genes, *AhCDPK2/-25/-28/-45*, exhibited a peak expression level at 3 h, which may suggest that these genes play a crucial role in mediating the plants’ early response to low Ca conditions.

## 4. Discussion

As the critical sensors and decoders of calcium signals, CDPKs are essential in the regulation of plant growth, development, and stress tolerance [4,5,9,12]. In the previous study, CDPKs have been identified in *Arabidopsis thaliana*, rice, maize, wheat, soybean, cotton, chickpea, *Medicago truncatula*, peach, and pineapple [6,17,21,22,23,24,25,26,27]. However, the functions of CDPKs are still poorly understood in cultivated peanut. In this study, a total of 45 putative *CDPK* genes in cultivated peanut were identified using a combination of BLASTP analysis, HMM analysis, and conserved domain verification (Figure 1, Appendix A). We conducted evolution analysis to reveal the gene expansion and the genetic relationship of the gene family (Figure 2 and Figure 3), which was supported by phylogenetic analysis. Phylogenetic analysis also classified 45 AhCDPKs into 4 subgroups based on their similarity (Figure 4). We further examined the structural features of the gene family, such as protein-conserved motifs, gene structure, and protein tertiary structures (Figure 5, Figure 6 and Figure 7), and found that they were consistent within each subgroup. To explore the potential function of the gene family, we predicted the regulatory mechanisms involving AhCDPKs genes and proteins. We analyzed the *cis*-regulatory elements and miRNA targets of *AhCDPKs* genes (Figure 8 and Figure 9), and the protein interaction network and functional annotations (GO and KEGG) of AhCDPKs proteins (Figure 10 and Figure 11). We discovered that the gene family had important roles in growth and development, stress response, and nutrient balance. Through expression analysis, we observed that *AhCDPK* genes had diverse expression patterns in different developmental stages (Figure 12). Moreover, different *AhCDPK* genes showed specific responses to various stresses, such as *AhCDPK5/-7/-18*, which were involved in cold, drought, and salt stress (Figure 13). Finally, we suggested that *AhCDPK* genes might have adaptive functions in peanut plants under Ca deficiency and that *AhCDPK2/-25/-28/-45* could serve as indicators for pod Ca nutrition (Figure 14).

The peanut is an intriguing allotetraploid species with two subgenomes, labeled A and B. These subgenomes are genetic inheritances from two ancestral species: *A. duranensis* (AA) and *A. ipaensis* (BB) [28,29,30]. Our research has revealed that 22 *CDPK* genes are present in *A. duranensis*, while 23 *CDPK* genes are found in *A. ipaensis* (Appendix A). Notably, previous research identified 35 *CDPK* genes in each of these wild peanuts, but only through the BLASTP method [59]. This implies that our approach is more effective. In contrast to the distribution pattern seen in the diploid parents, the cultivated peanut presents a different scenario: 23 *CDPK* genes are in the A-subgenome and 22 *CDPK* genes are in the B-subgenome (Figure 1 and Figure 2). This is largely due to the fact that the peanut subgenomes have evolved asymmetrically over time [29]. The total number of *AhCDPK* genes is equal to the sum of the genes in the two diploid parents, which indicates a translocation occurrence during allopolyploidization. When compared to several other plant species, the cultivated peanut stands out with a higher number of *AhCDPK* genes, such as *Arabidopsis thaliana* (34), rice (29), chickpea (22), *Medicago truncatula* (24), peach (17), and pineapple (17) [6,21,22,23,27,60]. However, the peanut’s total still falls short when compared to tetraploid sea-island cotton (84) and hexaploid wheat (85) [17,25]. One significant factor contributing to the large number of *AhCDPK* genes in peanuts is the occurrence of WGD events, which were the main duplication events (Figure 2). It is also worth noting that transposable elements make up a significant portion of the assembled genome sequence—a staggering 74% [29]. Our study further revealed evidence for two tandem duplication events within the *AhCDPK* genes. Specifically, *AhCDPK18* and *AhCDPK19* on the Arahy.07 chromosome are tandem duplicates, with their respective paralogs *AhCDPK41* and *AhCDPK42* on the Arahy.17 chromosome representing another tandem duplication event. We found evidence of 20 WGD events between two homologous genes, while the remaining 13 WGD events were not. The expansion of *AhCDPK* genes in the peanut could have occurred primarily through WGD between two homologous genes, followed by tandem duplication, segmental duplication, and allopolyploidization from the two ancient species. However, newly duplicated genes often exhibit functional redundancy, which can lead to gene loss over time [61]. To counteract this, mechanisms such as expression reduction come into play to facilitate the retention of duplicate genes and preserve their ancestral functions [61]. In our study, we observed that 12 *AhCDPK* genes in Cluster B displayed low expression levels in most of the 22 peanut tissues under normal conditions (Figure 12). This finding aligns with similar results reported in other gene families within the peanut genome [62,63,64]. These observations support the speculation that reduced expression may be advantageous in maintaining duplicate genes and their functional redundancy. The observation of more orthologs between peanut and *Arabidopsis* than between peanut and rice suggests that the expansion of the CDPK gene family in peanut is dependent on evolutionary processes (Figure 3). Orthologs are genes that have evolved from a common ancestor in different species and have retained similar functions [65]. In this case, the presence of more orthologs between peanut and *Arabidopsis* indicates a closer evolutionary relationship and a higher degree of conservation in the CDPK gene family.

Throughout the process of evolution, the CDPK gene family has exhibited a significant level of structural preservation across different plant species, ranging from mosses to angiosperms [4,10,12]. The representative function of CDPK is Ca^2+^-sensing functions and kinase activity. After the analysis of the MEME motif, Pfam, SMART, and PROSITE profiles, all 45 AhCDPK proteins included PKD and CaMLD (Figure 5, Appendix A). GO and KEGG analysis further confirmed the function of AhCDPKs (Figure 11). The VNTD and JD were also recognized by MEME. Like those in *Arabidopsis*, rice, maize, cotton, and peanut have CDPKs with less than four EF-hands. Only *AhCDPK41* has three EF-hands. There was a tandem duplication between *AhCDPK41* and *AhCDPK42*. Variable deletions were frequent in proximal chromosome regions [29]. *AhCDPK41* and *AhCDPK42* were in proximal Arahy.17. Thus, a variable deletion may come up during duplication.

In our study, we classified the CDPK gene family in peanut into four subgroups based on their general classification in *Arabidopsis* and rice [6,22]. However, in some species, such as sweet potato, there may be five subfamilies [66]. Interestingly, we observed that not every protein in the peanut CDPK gene family had homologous genes distributed on subchromosomes A and B. For example, AhCDPK3 did not have any paralogs. Additionally, the two pairs of homologous genes, AhCDPK20 and AhCDPK29, AhCDPK9 and AhCDPK40, were not located on the same subchromosomes, which was unexpected. To further support our findings, we analyzed the exon–intron structure and tertiary structures of the AhCDPK proteins (Figure 6 and Figure 7). These analyses provided additional evidence for the phylogenetic relationship among the AhCDPKs. We observed that most paralogs shared the same template protein, except for AhCDPK11 and AhCDPK33, as well as AhCDPK16 and AhCDPK39. The differences in template proteins between AhCDPK11 and AhCDPK33 may be attributed to the number of introns, resulting in different gene structures. AhCDPK33 had two additional introns compared to AhCDPK11, including an intron with a length of 8542, which was the longest among all AhCDPKs. While the exon-intron structure of AhCDPK16 and AhCDPK39 appeared similar, the reasons for the differences in their template proteins remain unclear and require further investigation. 

Most of the homologous paralogs exhibited similar biochemical properties, which were dependent on the subgroup classification [4]. In terms of pI, AhCDPK proteins in Subgroup I were found to be acidic, with pIs below 5.8 (Appendix A), similar to green algae and early land plants [4]. Conversely, AhCDPK proteins in subgroup IV were alkaline, with pIs beyond 9.1 (Appendix A). However, unlike other land plants, the pIs of AhCDPK proteins in Subgroup II varied, ranging from 5.31 to 8.2 (Appendix A). Specifically, both AhCDPK17 and AhCDPK38 in Subgroup II had a pI of 8.2. Only the above two paralogs lost the N-palmitoylation site in Subgroup II. The other proteins that lost N-terminal myristoylation or palmitoylation site were in Subgroup I. The specific functional implications of these variations in pI and post-translational modifications are still not fully understood and require further understanding.

The cultivated peanut is a vital crop with significant economic importance, particularly for oil production. However, it is susceptible to various abiotic stresses, including cold, drought, and salt [31]. Understanding and analyzing the *CDPK* genes in peanut can have profound implications for enhancing stress responses in this crop. As shown in Figure 8, the presence of ABA-responsive elements (ABRE, ABRE3a, and ABRE4), drought-responsive elements (MBS, Myb, MYB, Myb-binding site, MYB-like sequence, Myc, and MYC), and cold-responsive elements (LTR) suggests the potential involvement of *AhCDPKs* in regulating multiple responses to environmental stresses. Furthermore, the predicted gene functions of AhCDPKs were further supported by the GO enrichment analysis, as depicted in Figure 11. This analysis indicated the role of *AhCDPKs* in stress response mechanisms. Figure 13 illustrates that the application of salt, drought, and cold stresses resulted in the differential expression of 7, 17, and 17 AhCDPK genes, respectively. The variation in the number of genes can be attributed to the different time points at which the stress treatments were applied. In this study, drought and cold stresses were examined at three different time points, while salt stress was examined at only one time point due to limited public RNA-seq data. It is worth noting that previous research has shown the potential of CDPK genes in enhancing stress tolerance [17,18,19,20,21,25]. For instance, heterogeneous expresses *SiCDPK24*, *TaCDPK25-U-AS1*, *TaCDPK25-U-AS2,* or *GmCDPK3* in transgenic *Arabidopsis* lines displayed enhanced tolerance to drought stress [17,18,19]. Additionally, the *Arabidopsis cpk23* mutant showed increased tolerance to drought and salt stresses, while *AtCPK23* overexpressing plants demonstrated reduced resistance to these stresses [67]. Furthermore, in rice, the induction in *OsCDPK7* by cold and salt stresses, as well as its overexpression, led to increased tolerance to cold, salt, and drought [16]. These findings highlight the role of CDPKs in stress responses and their potential applications in crop improvement. To further explore the potential functions of AhCDPKs, protein interaction prediction was performed. This analysis can provide insights into the interactions and networks in which AhCDPKs may participate, shedding light on their role in stress signaling pathways. In our study, we found that all 32 AhCDPK proteins interacted with 5 peanut RBOH proteins, as shown in Figure 10. It is known that CDPKs can activate NADPH oxidase through phosphorylation, playing a crucial role in stress-induced oxidative damage [11,12]. In rice, for example, *OsCPK12* regulates ROS homeostasis under salt stress by inducing the expression of the ROS-scavenging gene *OsAPX2/OsAPX8* and inhibiting the NADPH oxidase gene *OsRBOHI* [68]. Interactions between CDPKs and other proteins have also been observed in other plant species. For example, in wheat, TaCDPK13 interacts with TaNOX7 to enhance ROS production, playing a crucial role in drought tolerance [69]. In peach, the interaction between PpCDPK7 and PpRBOH is believed to be the intersectional point of calcium and ROS signal transmission during cold storage of peach fruits [21]. This suggests that *AhCDPKs* may also participate in ROS signaling pathways in peanut in response to environmental stress. Based on these findings, it is reasonable to speculate that *AhCDPKs* in peanut may also participate in ROS signaling pathways in response to environmental stress. Further research is needed to elucidate the specific mechanisms and functions of *AhCDPKs* in stress tolerance and ROS signaling pathways in peanut.

Early studies have indicated the involvement of *AhCDPK* in peanut fruit development under Ca deficiency, and RNA-seq analysis has suggested their role in pod development [33,34,36,58]. In our research, we have found evidence supporting the important role of *AhCDPKs* during peanut fruit and pod development, particularly under Ca-deficient soil. The identification of the RY-element, a seed-specific regulation element, in Figure 8 suggests that the *AhCDPK* genes may be involved in seed development. The GO analysis results, which imply the important role of AhCDPKs in the regulation of cellular processes, further support the involvement of these genes in seed development (Figure 11). Significant advancements have been made in identifying the targets of peanut miRNAs involved in various developmental processes [70,71,72]. In the early stages of peanut pod development, a comprehensive study revealed the presence of 70 known and 24 novel miRNA families [71]. It was observed that almost all miRNAs-targeting *AhCDPK* genes were specifically expressed during the pod development stage, with the exception of ahy-miR3510. Another study focused on early peanut embryos and identified 29 known and 132 potential novel miRNAs, including all miRNAs-targeting *AhCDPK* genes except ahy-miR3520-3p and ahy-miR3520-5p [72]. Remarkably, among the AhCDPK genes upregulated under Ca deficiency, 12 out of the 26 identified genes were found to be targeted by miRNAs. This suggests that miRNAs play a role in modulating the expression of *AhCDPK* genes and potentially influencing seed development outcomes, particularly in response to Ca deficiency. The protein interaction network analysis indicating the involvement of AhCDPKs in ROS signaling pathways suggests a potential link between miRNA-mediated regulation of pod development and ROS signaling (Figure 10). Further investigation is needed to clarify whether miRNAs influenced pod development by affecting ROS signaling pathways in peanut. Additionally, Figure 14A shows that 26 *AhCDPKs* were upregulated in response to Ca deficiency, with 12 of them belonging to Cluster A (accounting for 71%) and 9 belonging to Cluster C (accounting for 56%). This indicates that approximately 80% of the upregulated *AhCDPKs* were expressed at moderate levels in most peanut tissues, with 46% being highly expressed, particularly during pod development. These findings provide additional evidence supporting the importance of *AhCDPKs* during pod development, consistent with early research [34,58]. Furthermore, we conducted a detailed investigation into the role of *AhCDPKs* by analyzing the expression levels of eight significantly upregulated *AhCDPKs* in the seedling stage under Ca deficiency treatment (Figure 14). Among these, four genes (*AhCDPK2/-25/-28/-45*) were identified as potential signals of Ca nutrients during the earlier stages of development. Notably, *AhCDPK2* and *AhCDPK25* were paralogs in Subgroup III and were homologous to *AtCPK10/-30* in *Arabidopsis*. In *Arabidopsis*, Subgroup III *AtCPK10/-30-32* are master regulators that orchestrate primary nitrate responses [73]. Nitrate–CPK signaling phosphorylates conserved NIN-LIKE PROTEIN (NLP) transcription factors to specify downstream gene reprogramming [73]. However, as peanut are legume crops and have a complex process of nitrogen absorption due to their symbiotic relationship with rhizobia, it is possible that there are some changes in the mechanisms involved in nutrient signaling. Further research is needed to elucidate the specific downstream targets and regulatory mechanisms of *AhCDPKs* in response to Ca deficiency.

## 5. Conclusions

We identified 45 CDPK genes with high confidence in the genomes of cultivated peanut. These genes were categorized into four subgroups based on a phylogenetic analysis. Various aspects such as protein physiological properties, protein tertiary structure, and conserved motifs were dependent on subgroup classification. Duplication events were identified that occurred in the peanut genome by collinearity analysis. Cis-acting regulatory elements in promoters, protein interaction networks, and GO and KEGG annotation evaluations implied the function of AhCDPK proteins in environmental stimuli. RNA-seq data from 22 different tissues revealed that *AhCDPK* genes in Cluster A and Cluster C play significant roles in plant development. The expression profile analysis of *AhCDPKs* due to various stresses implied their crucial roles in response to multiple signaling pathways. Interestingly, we observed a significant upregulation of *AhCDPK* genes in response to Ca deficiency. This observation was further supported by the RT-qPCR analysis in seedlings, highlighting the crucial involvement of *AhCDPKs* in peanut Ca nutrient regulation. By conducting a genome-wide identification of CDPKs in peanut, our study has the potential to significantly enhance our understanding of the genetic basis underlying peanut growth, development, and stress resistance.

## Figures and Tables

**Figure 1 cells-12-02676-f001:**
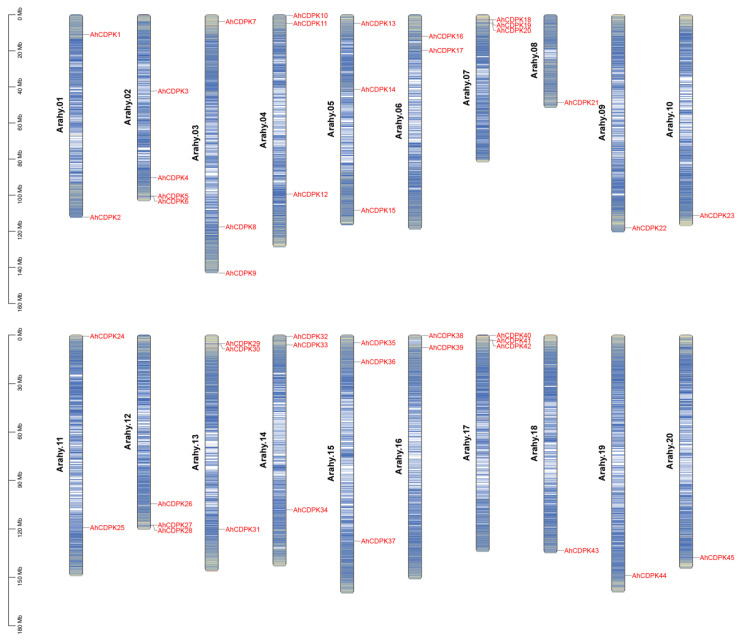
Chromosome density and chromosomal distribution of the *AhCDPK* genes. Chromosome numbers are provided at the left of each chromosome.

**Figure 2 cells-12-02676-f002:**
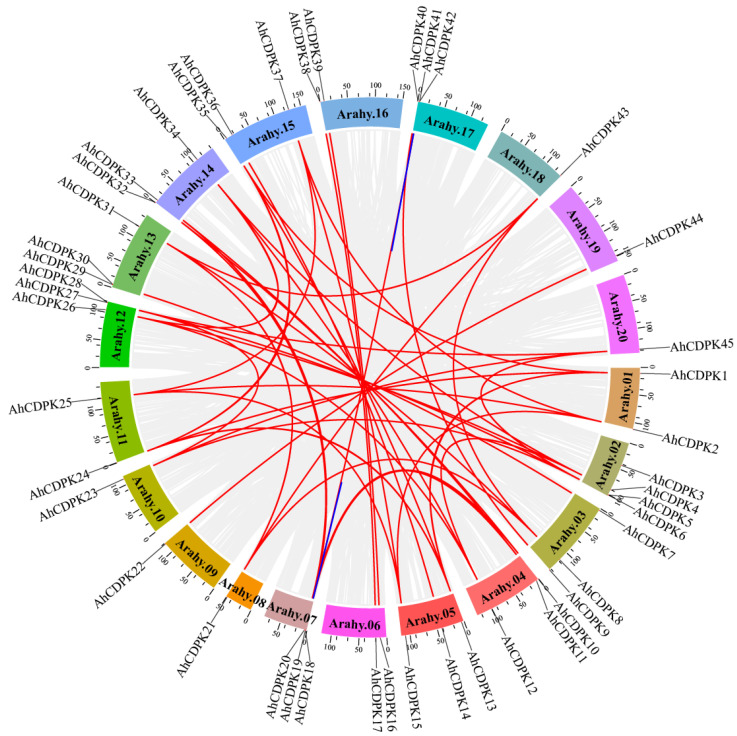
Synteny relationship of *AhCDPKs* in cultivated peanut. The blue lines indicate tandem duplicated gene pairs, and the red lines indicate segmentally duplicated and WGD gene pairs. The gray lines indicate segmentally duplicated gene pairs within the peanut genome. The scale bar marked on the chromosome indicates chromosome lengths (Mb).

**Figure 3 cells-12-02676-f003:**
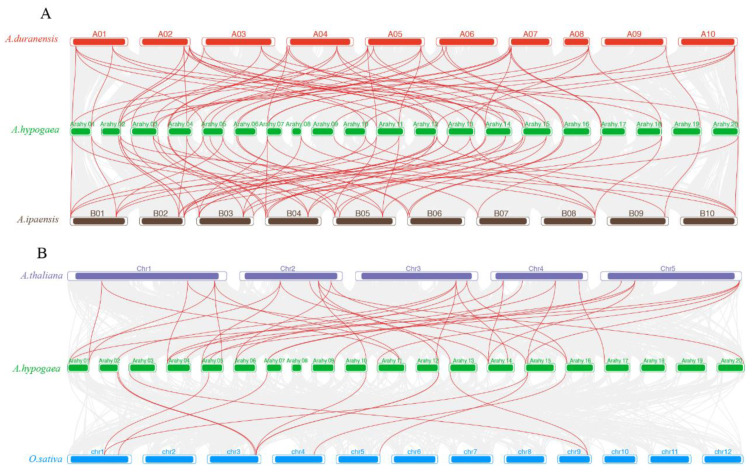
Synteny analysis of *CDPK* genes between cultivated peanut and its two diploid ancestors (**A**) and among peanut, rice, and *Arabidopsis* (**B**). Gray lines in the background indicate the collinear blocks within peanut and other plant genomes, while the red lines highlight the syntenic *CDPK* gene pairs.

**Figure 4 cells-12-02676-f004:**
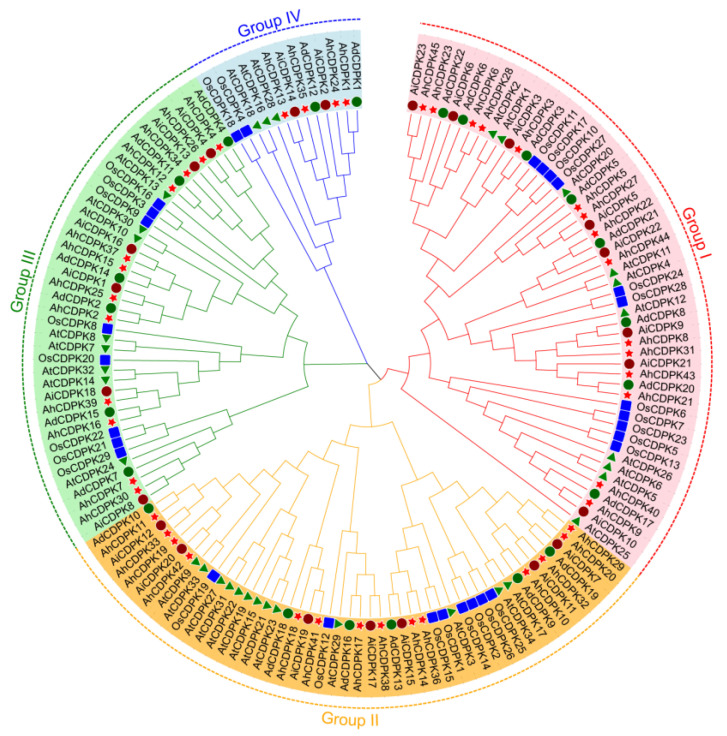
Phylogenetic analysis of CDPK proteins. The proteins of peanut, *Arabidopsis*, rice, *Arachis duranensis*, and *Arachis ipaensis* are represented in red star, green triangle, blue square, dark green circle, and dark red circle, respectively.

**Figure 5 cells-12-02676-f005:**
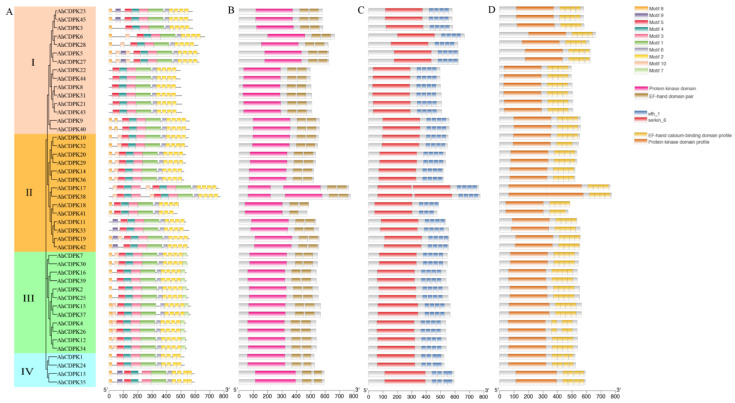
Conserved motifs analyses of AhCDPK proteins. (**A**) Conserved motif by MEME. The colored boxes on the right denote motifs 1–10. (**B**) Pfam motif. (**C**) SMART motif. (**D**) PROSITE profile.

**Figure 6 cells-12-02676-f006:**
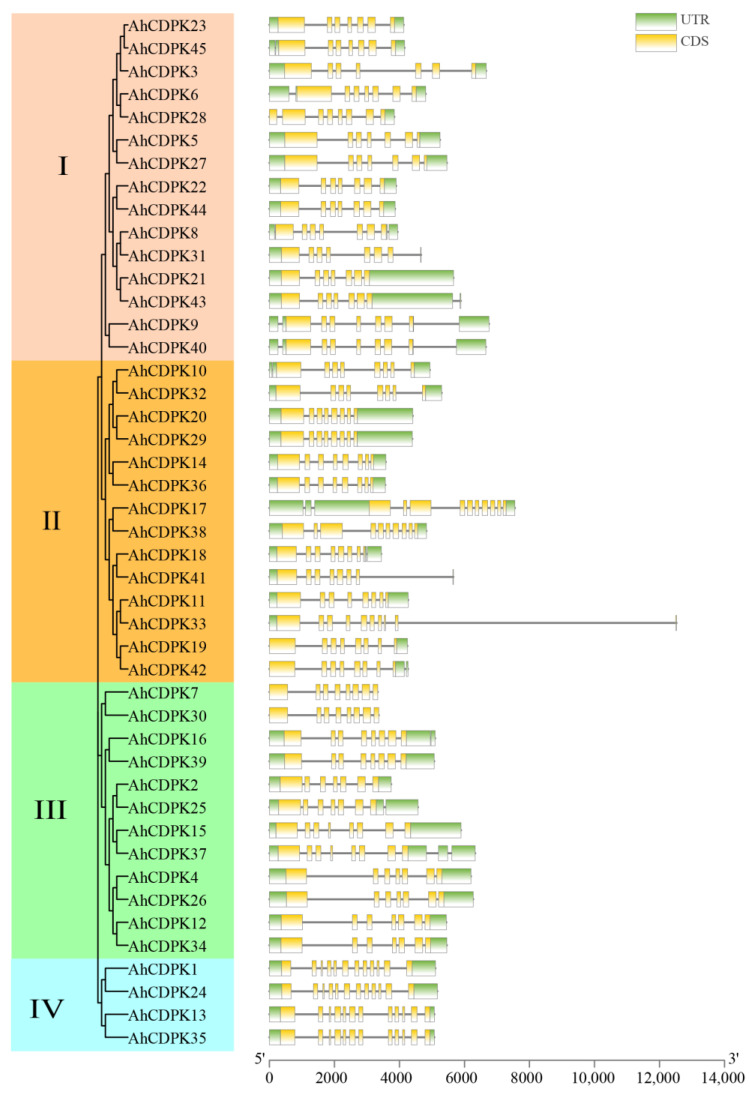
Gene structure analyses of *AhCDPK* genes. The green boxes, black lines, and yellow boxes represent UTR, introns, and CDS, respectively.

**Figure 7 cells-12-02676-f007:**
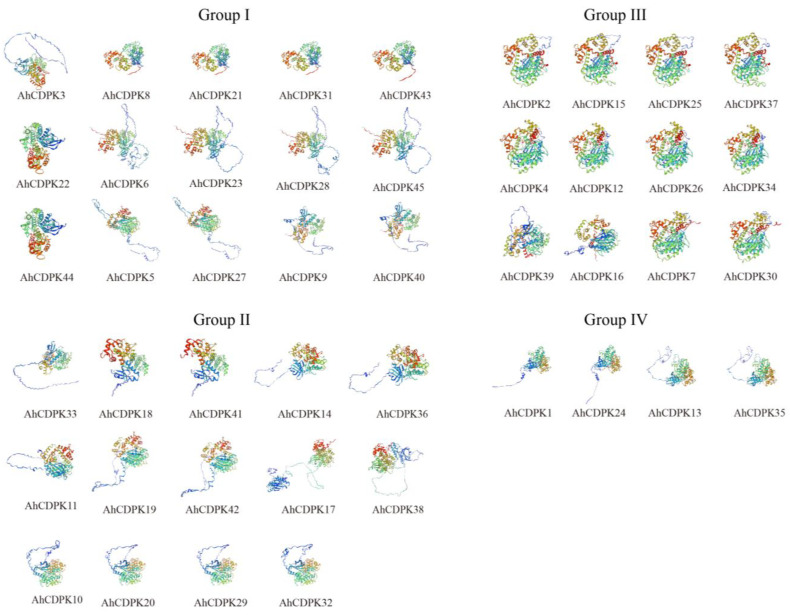
Prediction of AhCDPK proteins’ tertiary structure by AlphaFold2 method. Models were visualized using rainbow colors from N to C terminus.

**Figure 8 cells-12-02676-f008:**
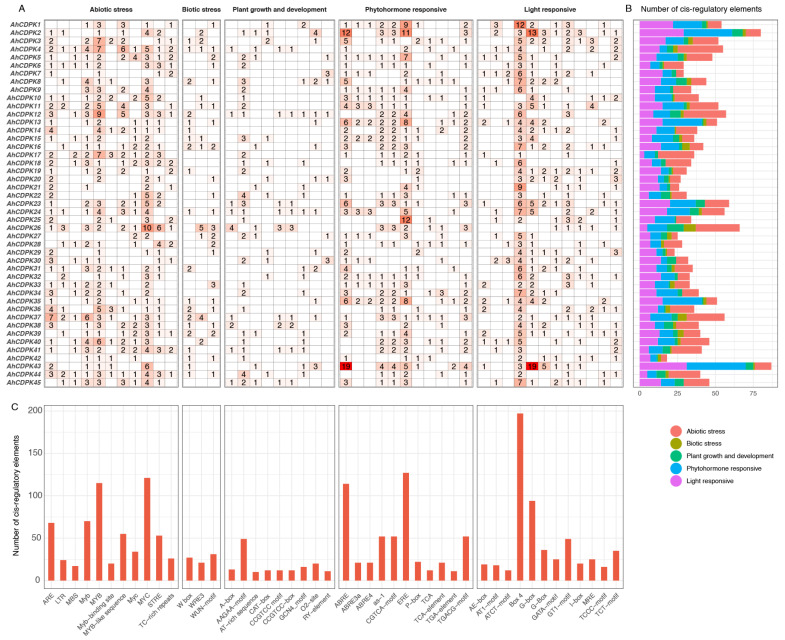
*Cis*-regulatory elements analysis of *AhCDPK* genes upstream regions. (**A**) The numbers and the depth of red represent the frequency of the elements that occur in the promoter region. (**B**) The statistics of categories on every *AhCDPK*. Different categories are present with different colors. (**C**) The number of each identified element.

**Figure 9 cells-12-02676-f009:**
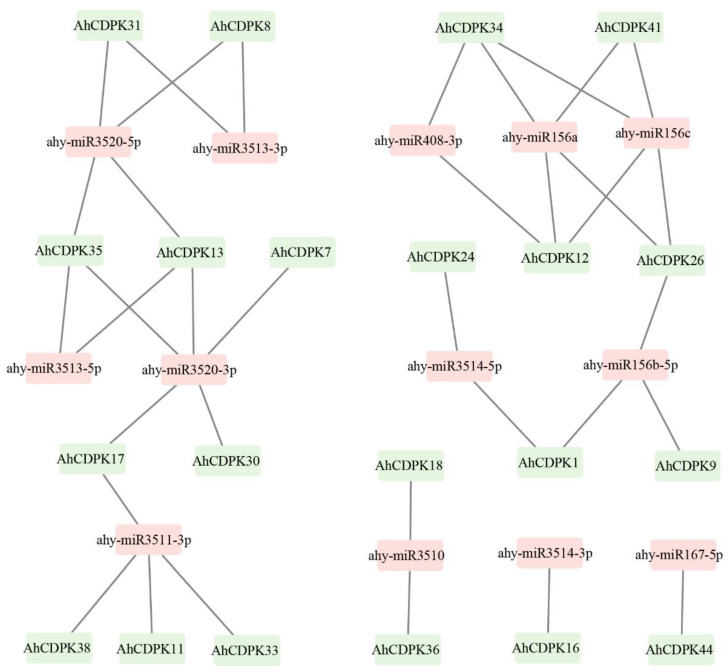
Network map of projected miRNA targeting *AhCDPK* genes. The green boxes correspond to *AhCDPK* genes, and the pink boxes indicate predicted miRNAs.

**Figure 10 cells-12-02676-f010:**
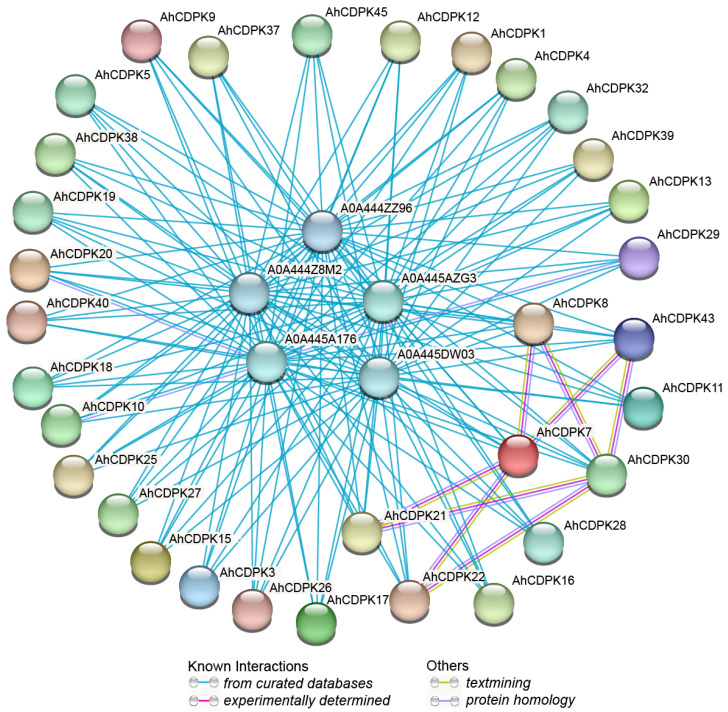
Predictive interaction network of CDPK proteins in peanut. Network nodes represent proteins, and edges represent protein–protein associations.

**Figure 11 cells-12-02676-f011:**
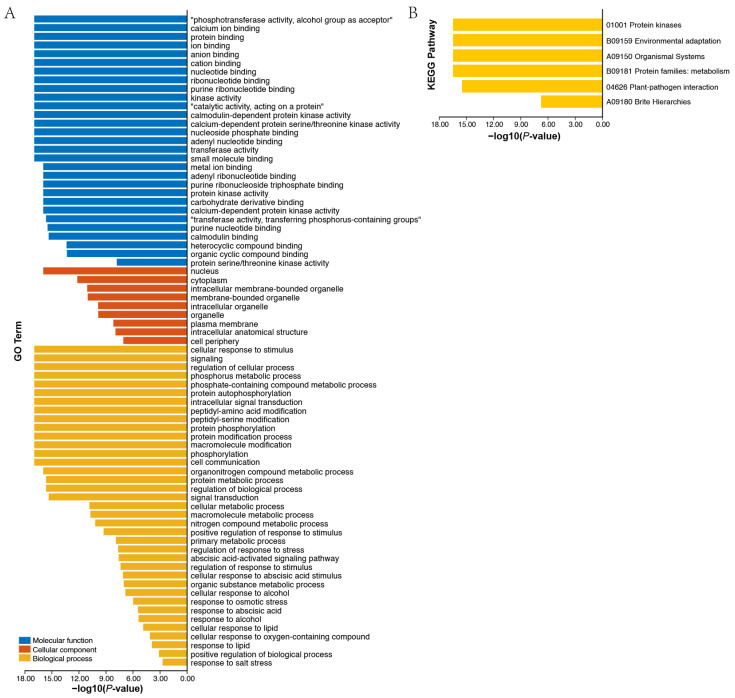
GO (**A**) and KEGG (**B**) enrichment analysis of *AhCDPKs*.

**Figure 12 cells-12-02676-f012:**
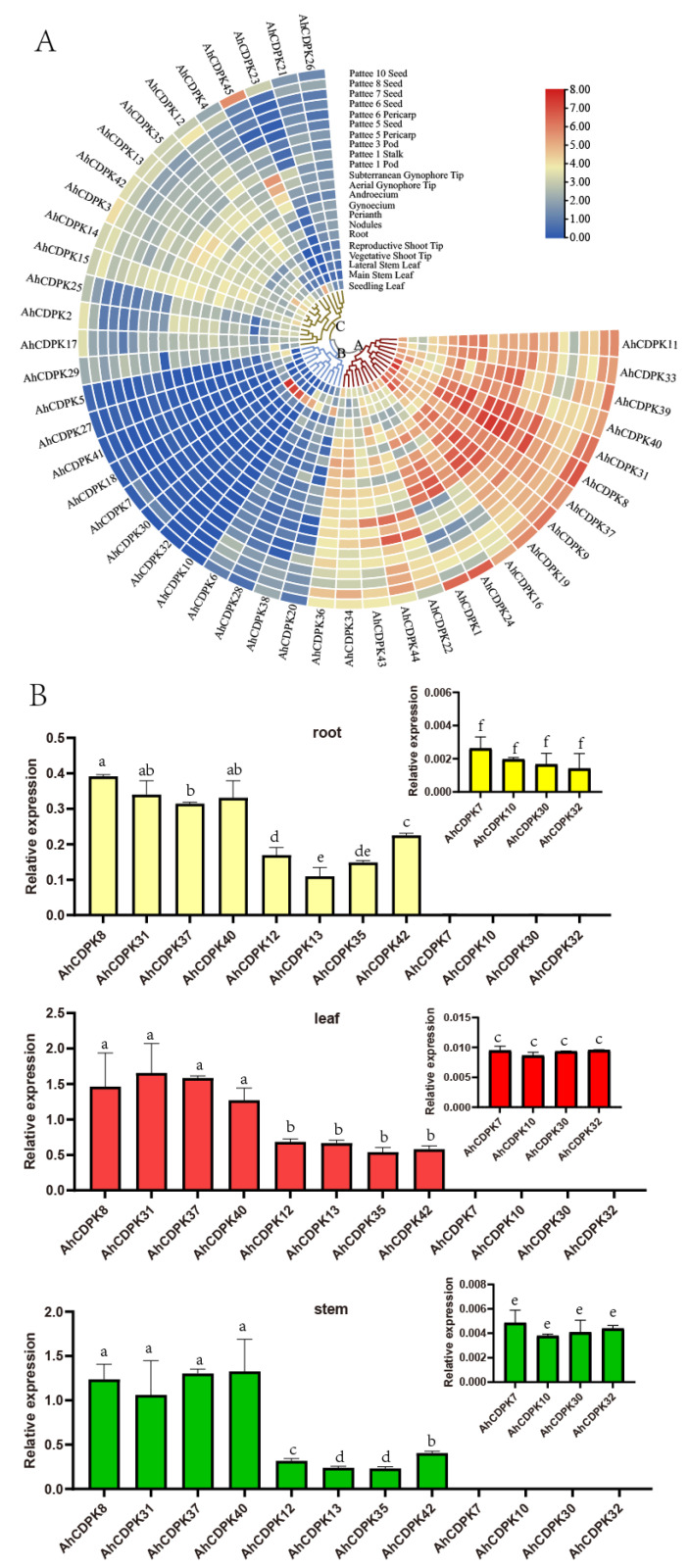
Expression patterns of *AhCDPK* genes in different developmental tissues. (**A**) RNA-seq analysis. The heatmap was constructed using the mean of 1, 2, or 3 biological replicates, details in Appendix A. The labels A, B, and C in the heatmap were shortened to indicate Cluster A, Cluster B, and Cluster C, respectively. (**B**) RT-qPCR analysis. Data are means ± SD (n = 6). Different letters indicate significant differences (Duncan’s test, *p* < 0.05) in each tissue including the subgraph.

**Figure 13 cells-12-02676-f013:**
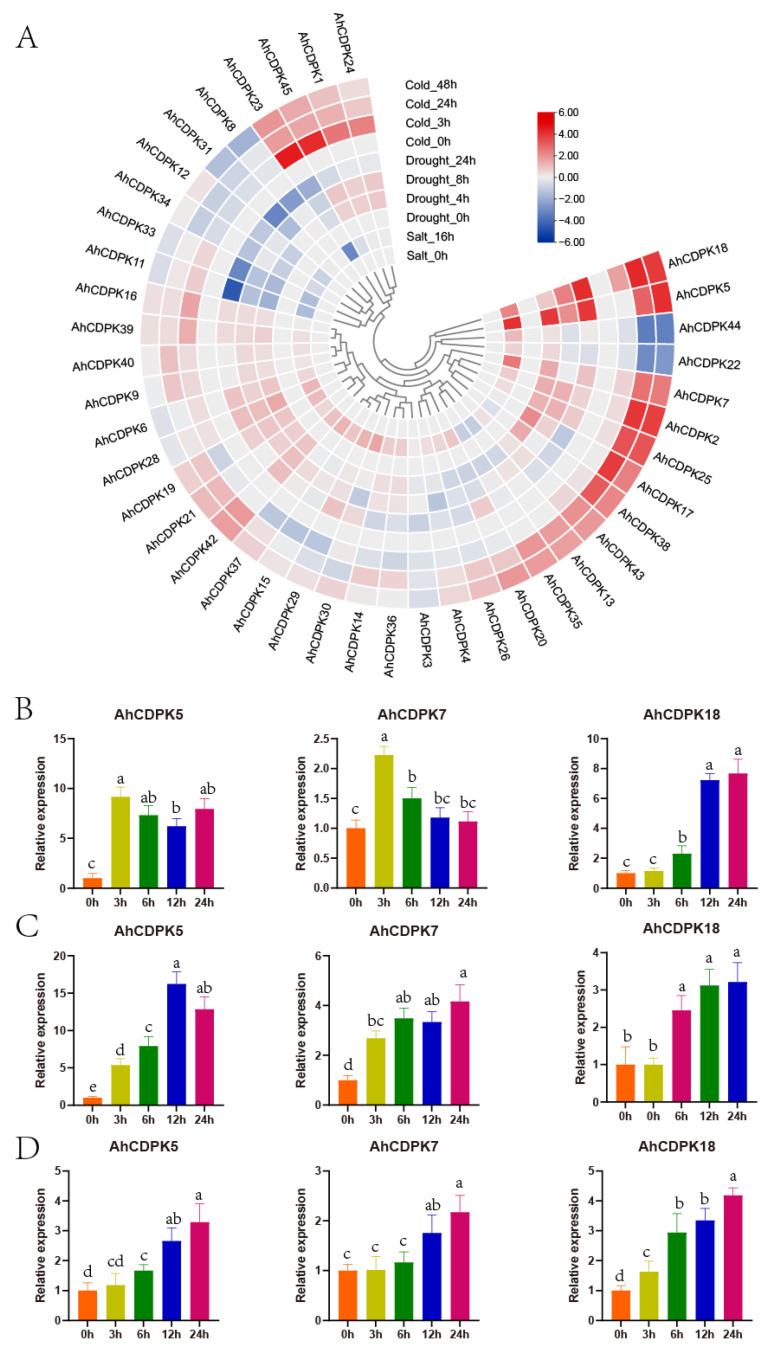
Expression patterns of *CDPK* genes in peanut in response to abiotic stresses. (**A**) RNA-seq analysis. The heatmap was constructed using the mean of three biological replicates. RT-qPCR analysis when exposed to drought (**B**), salt (**C**), and cold (**D**). Data are means ± SD (n = 6). Different letters indicate significant differences (Duncan’s test, *p* < 0.05).

**Figure 14 cells-12-02676-f014:**
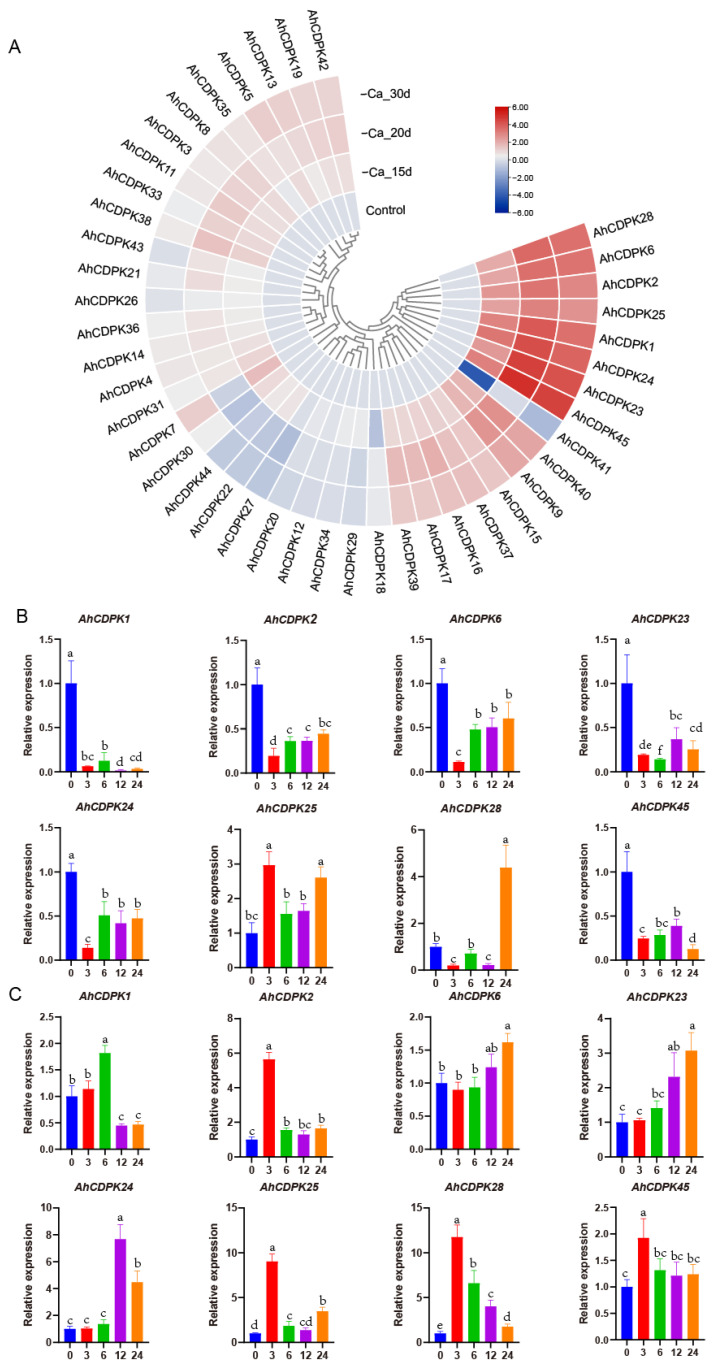
Expression patterns of *CDPK* genes in peanut pods in response to Ca deficiency. (**A**) RNA-seq analysis. The heatmap was constructed using the mean of three biological replicates. RT-qPCR analysis in (**B**) leaf and (**C**) root. Data are means ± SD (n = 6). Different letters indicate significant differences (Duncan’s test, *p* < 0.05).

## Data Availability

Data are contained within the article or Appendix A.

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
