# Peer review of "Genome-Wide Identification and Characterization of CDPK Gene Family in Cultivated Peanut (Arachis hypogaea L.) Reveal Their Potential Roles in Response to Ca Deficiency"

_cells, 2023, doi:10.3390/cells12232676_

Round 1

Reviewer 1 Report

Comments and Suggestions for Authors

The work is very complete in terms of the identification of the CDPK gene family in peanut, its characterization, genomic evolution, gene structure, motifs present in each gene, regulatory elements, its possible interaction with miRNA and other proteins.

However, I consider that the most relevant aspect of this work is the identification of those AhCDPK genes related to abiotic stress, their tissue-specific expression, very particularly with the definition of those AhCDPK genes (2, 25, 28 and 45) directly associated to Ca deficiency in roots. These genes are excellent candidates for further understanding of Ca signaling mechanisms and their role in plant developmental processes.

Given the above, it seems to me to be a work that enriches knowledge related to the perception mechanisms of plants and should be published.

I do think it is important to improve the definition of the figures. These contain a large number of data, and it would be best to present only those elements of the figure that are most relevant to the field of study.

The work is very complete in terms of the identification of the CDPK gene family in peanut, its characterization, genomic evolution, gene structure, motifs present in each gene, regulatory elements, its possible interaction with miRNA and other proteins.

However, I consider that the most relevant aspect of this work is the identification of those AhCDPK genes related to abiotic stress, their tissue-specific expression, very particularly with the definition of those AhCDPK genes (2, 25, 28 and 45) directly associated to Ca deficiency in roots. These genes are excellent candidates for further understanding of Ca signaling mechanisms and their role in plant developmental processes.

Given the above, it seems to me to be a work that enriches knowledge related to the perception mechanisms of plants and should be published.

I do think it is important to improve the definition of the figures. These contain a large number of data, and it would be best to present only those elements of the figure that are most relevant to the field of study.

Author Response

Comments 1: I do think it is important to improve the definition of the figures. These contain a large number of data, and it would be best to present only those elements of the figure that are most relevant to the field of study.

Response 1: Thank you for pointing this out. I agree with this comment. Therefore, I have changed the definition of Figure 1-2 and Figure 6-14 and checked all the definition to keep the concision.

4. Additional clarifications

This research was also funded by Shandong Key R&D Program (Major Scientific and Technological Innovation Project ZFJH202310).

Reviewer 2 Report

Comments and Suggestions for Authors

The research article entitled “Genome-Wide Identification and Characterization of CDPK 2

gene family in Cultivated Peanut (Arachis hypogaea L.) Reveal 3 Their Potential Roles in Response to Ca Deficiency” provides the overall information about the CDPK genes in peanut (Arachis hypogaea L.) which is one of the important crops.   However, I think the manuscript can be improved before publication.

1.   Introduction: This is well written, and the objective is quite clear.

2.   Methods:

2.1  Please add information on how to prepare Ca free solution for plant growth. Please also indicate experimental design, number of replication and statistical analysis for gene expression comparison.

2.2  Please indicate number of replications used for RNA-Seq analysis

3.   Results:

3.1  Increase resolution for all figures.

3.2  For figure 1, indicate the chromosome number of each chromosome.

3.3  The connection between the results in each analysis should be added to the research and clear or strong hypothesis should be made for the expression analysis. It seems that the transcriptome data in Ca deficiency condition was the only information used for gene selection for qualitative PCR analysis.

3.4  Are there any control for the gene expression when plants were grown in normal condition? Please show the evidence and indicate in the materials and methods.

4. Discussion: please provide the connection between each analysis and how these data help to elucidate or support the finding in genes responsible for Ca deficiency.

Author Response

Comments 1: Please add information on how to prepare Ca free solution for plant growth. Please also indicate experimental design, number of replication and statistical analysis for gene expression comparison.

Response 1: Agree. Thanks for your suggestion. I apologize for any misunderstanding caused by my oversight. Information about Ca free solution have been added in the materials and methods. Experimental design, number of replication and statistical analysis for gene expression comparison were also revised in materials and methods in line 198-206.

Comments 2: Please indicate number of replications used for RNA-Seq analysis

Response 2: Thank you for pointing this out. The replications of all RNA-Seq analysis were indicated in line 181-183 and in Table S1 in Additional file S2.

Comments 3: Increase resolution for all figures.

Response 3: Agree. I have enhanced the resolution of all figures included in the study. Each figure has been rendered in a vector style, ensuring high-quality and scalable images. Comments 4: For figure 1, indicate the chromosome number of each chromosome.

Response 4: Agree. I have modified figure 1 and the chromosome number of each chromosome were on the left marked as black.

Comments 5: The connection between the results in each analysis should be added to the research and clear or strong hypothesis should be made for the expression analysis. It seems that the transcriptome data in Ca deficiency condition was the only information used for gene selection for qualitative PCR analysis.

Response 5: Agree. Thank you for the suggestion. Thank you for the suggestion. I have performed RT-qPCR experiments in these days to confirm our hypothesis in the section of spatial expression analysis of AhCDPKs and expression analysis of AhCDPKs under abiotic stresses. I also have revised in the results according to your suggestion. In each analysis, I have added the connection between the results.

Comments 6: Are there any control for the gene expression when plants were grown in normal condition? Please show the evidence and indicate in the materials and methods.

Response 6: Agree. I apologize for any confusion. Non-treated samples at 0 hours were served as the control. Accordingly, I have revied the materials and methods in line 201-206.

Comments 7: Discussion: please provide the connection between each analysis and how these data help to elucidate or support the finding in genes responsible for Ca deficiency.

Response 7: Agree. Thank you for the suggestion. I have changed the discussion according to your suggestion in line 616-632 and 760-779.

4. Additional clarifications

This research was also funded by Shandong Key R&D Program (Major Scientific and Technological Innovation Project ZFJH202310).

Reviewer 3 Report

Comments and Suggestions for Authors

Article “Genome-wide ….. Ca Deficiency” discussed about the identification of 45 calcium-dependent protein kinase (CDPK) genes in peanut from existing gene data bank using in silico approach with Arabidopsis CDPK genes. They further analyze and characterize genes using computational biology and bioinformatics. As a wet-lab experiment, they checked expression analysis using qRT-PCR. The study is solely based on bioinformatics and data-mining. The study seems good but more of bioinformatics type. If it falls under the aim and scope of journal (Cells) and may accepted for the publication in its present form. However, a melt-curve analysis of qRT-PCR (for all genes) may provide as supplementary just to confirm the primer-suitability and any other unspecific binding.

Author Response

Comments 1: However, a melt-curve analysis of qRT-PCR (for all genes) may provide as supplementary just to confirm the primer-suitability and any other unspecific binding.

Response 1: Thank you for pointing this out. I agree with this comment. Accordingly, I have added melt-curve analysis of qRT-PCR (for all genes) as Figure S1 in Additional file S1.

4. Additional clarifications

This research was also funded by Shandong Key R&D Program (Major Scientific and Technological Innovation Project ZFJH202310).